# Molecular analysis of human Ero1 reveals novel regulatory mechanisms for oxidative protein folding

Antti Moilanen⊙, Kati Korhonen, Mirva J Saaranen⊙, Lloyd W Ruddock⊙

**Oxidative protein folding in the ER is driven mainly by oxidases of the endoplasmic reticulum oxidoreductin 1 (Ero1) family. Their action is regulated to avoid cell stress, including hyperoxidation. Previously published regulatory mechanisms are based on the rearrangement of active site and regulatory disulfides. In this study, we identify two novel regulatory mechanisms. First, both human Ero1 isoforms exist in a dynamic mixed disulfide complex with protein disulfide isomerase, which involves cysteines (Cys166 in Ero1α and Cys165 in Ero1β) that have previously been regarded as being nonfunctional. Second, our kinetic studies reveal that Ero1 not only has a high affinity for molecular oxygen as the terminal acceptor of electrons but also that there is a high cooperativity of binding (Hill coefficient >3). This allows Ero1 to maintain high activity under hypoxic conditions, without compromising cellular viability under hyper-hypoxic conditions. These data, together with novel mechanistic details of differences in activation between the two human Ero1 isoforms, provide important new insights into the catalytic cycle of human Ero1 and how they have been fine-tuned to operate at low oxygen concentrations.**

## Introduction

Control of the redox homeostasis of the ER is of fundamental importance for the effective formation of native disulfide bonds. It is manipulated primarily by oxidative enzymes of the flavin-dependent endoplasmic reticulum oxidoreductin 1 (Ero1) family (Frand & Kaiser, 1998; Pollard et al, 1998; Cabibbo et al, 2000; Pagani et al, 2000) and by a glutathione buffer formed from a mixture of reduced glutathione (GSH) and oxidized glutathione (GSSG) in a molar ratio of 3:1–6:1 (Hwang et al, 1992; Bass et al, 2004; Dixon et al, 2008). Ero1 forms the main pathway for disulfide formation in the ER by transferring disulfides to protein disulfide isomerase (PDI) (Mezghrani et al, 2001; Appenzeller-Herzog et al, 2010). PDI has a dual role in native disulfide formation. In its oxidized state, it can transfer disulfides to nascent polypeptides, whereas in its reduced

state, it can shuffle non-native disulfides into the native state (isomerize) (reviewed in Hatahet & Ruddock [2009]). GSH ensures that a pool of PDI in the ER is kept in the reduced state to allow for catalysis of isomerization (Chakravarthi & Bulleid, 2004), whereas GSSG contributes to total disulfide pool by oxidizing PDI (Lappi & Ruddock, 2011). The main route for oxidation of PDI is via the action of Ero1. As a pool of PDI needs to be in the reduced state, the activity of Ero1 requires regulation. Regulation of Ero1 is also required as Ero1 generates disulfides de novo by reducing molecular oxygen into hydrogen peroxide (Gross et al, 2006; Wang et al, 2009); hence, uncontrolled disulfide bond formation would not only lead to hyperoxidation of the ER but also to the accumulation of cytotoxic reactive oxygen species. The mechanism for inactivation of Ero1 is based on the formation of regulatory disulfides (Appenzeller-Herzog et al, 2008). PDI has been suggested to modulate this regulatory switch in a feedback mechanism manner by possessing the capability to activate (Appenzeller-Herzog et al, 2008) and possibly inactivate (Shepherd et al, 2014; Zhang et al, 2014) Ero1. However, the exact mechanisms leading to inactivation of Ero1 are still unknown with auto-oxidation of Ero1 regulatory disulfides also reported (Zhang et al, 2014).

There are two Ero1 isoforms in the mammalian ER, Ero1α, and Ero1β. In contrast, many other eukaryotic species survive with only one Ero1 enzyme, for example, Ero1p in yeast. The reason for the need of two homologous Ero1 enzymes remains unanswered. The two isoforms are similar in molecular weight and amino acid distribution (Pagani et al, 2000) and contain a similar conserved cysteine distribution (Fig 1A). Their mechanism of action, therefore, is likely similar: both contain a flavin adenine dinucleotide (FAD) cofactor, which uses oxygen to oxidize an inner active site that oxidizes an outer active site, which in turn forms a transient mixed disulfide intermediate with an active site of PDI finally resolving into reduced Ero1 and oxidized PDI (Frand & Kaiser, 1999; Wang et al, 2009). Mixed disulfides between Ero1 and PDI have been reported for both isoforms (Benham et al, 2000; Mezghrani et al, 2001; Dias-Gunasekara et al, 2005; Appenzeller-Herzog et al, 2010), but no structural studies on the complex have been reported. Besides differences in tissue distribution (Pagani et al, 2000; Dias-Gunasekara et al, 2005), the need for two isoforms possibly lies in differential regulation. Although both have the same conserved cysteines for the primary regulatory switch

Faculty of Biochemistry and Molecular Medicine and Biocenter Oulu, University of Oulu, Oulu, Finland

Correspondence: Lloyd.Ruddock@oulu.fi

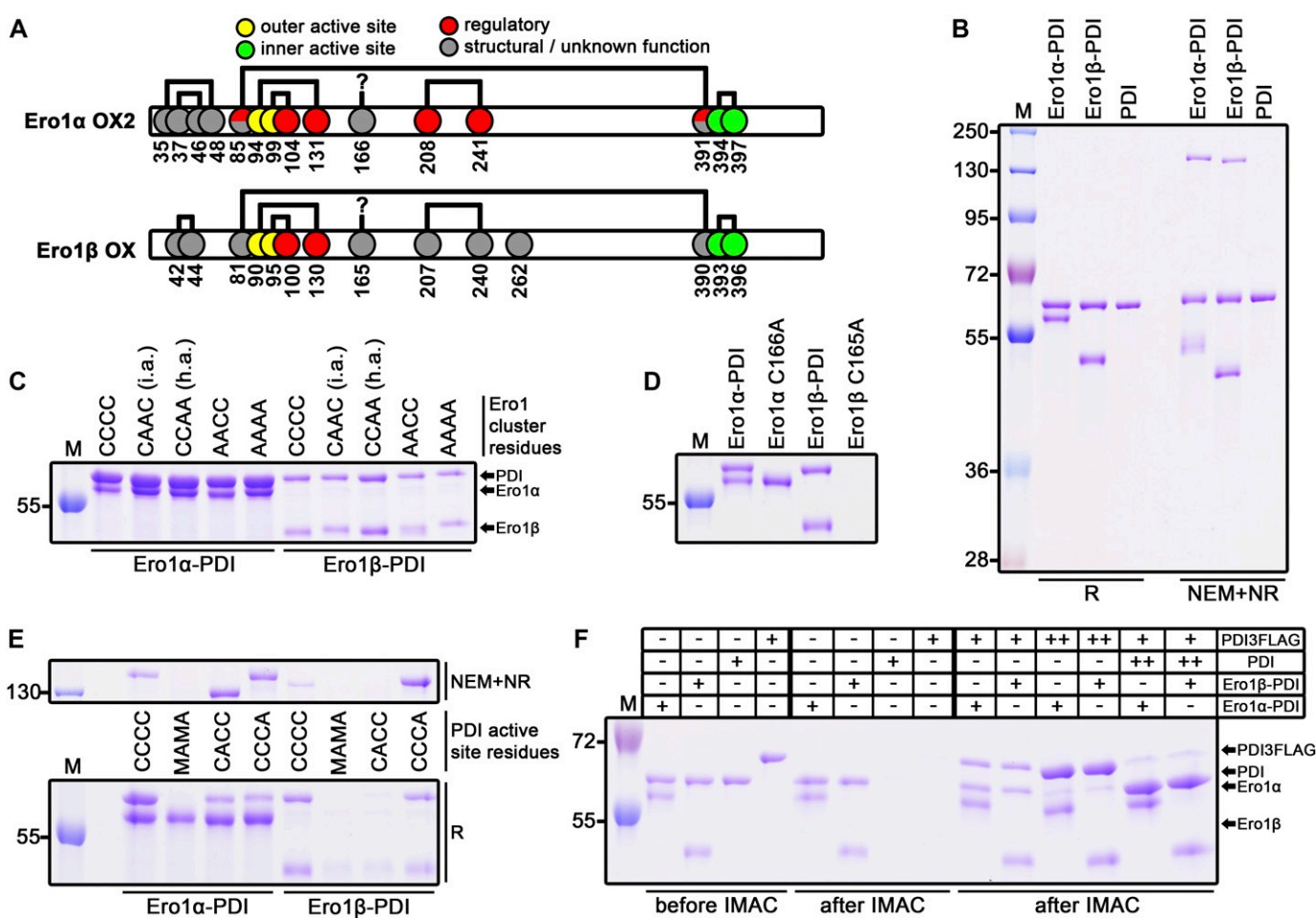

**Figure 1. Ero1 complex analysis.**
**(A)** Disulfide patterns of Ero1α OX2 (Appenzeller-Herzog et al, 2008; Inaba et al, 2010; Hansen et al, 2012) and Ero1β oxidized (OX) redox states (Hansen et al, 2014). Cysteines are shown as circles with reported functions indicated with colors. Disulfide connectivity is represented by black lines. Cys166 of Ero1α has previously been suggested to form a disulfide bond to an unknown partner (Appenzeller-Herzog et al, 2008). **(B)** SDS–PAGE analysis of purified wild-type Ero1α complex, wild-type Ero1β complex, and monomeric PDI. **(C)** Mixed disulfide between Ero1 and PDI was studied by introducing mutations to the outer active site and the adjacent regulatory cysteines of Ero1. CCCC, wild-type Ero1; CAAC (i.a.), inactivating Ero1α C99/104A or Ero1β C95/100A mutants; CCAA (h.a.), hyperactivating Ero1α C104/131A or Ero1β C100/130A mutants; AACC, outer active site Ero1α C94/99A or Ero1β C90/95A mutants; and AAAA, all four cysteines mutated. Samples were reduced by β-mercaptoethanol. **(D)** Reducing SDS–PAGE analysis of loss of mixed disulfide by mutating Cys166 of Ero1α or Cys165 of Ero1β to alanine. **(E)** PDI side of the mixed disulfide was analyzed by mutating active site cysteines (Cys53/55/397/400) to either alanine (A) or methionine (M) as indicated. The same protein samples were R or NEM+NR, run on different gels, and aligned. **(F)** Reducing SDS–PAGE analysis of exchange of PDI molecules in the Ero1 complexes. His-tagged Ero1 complexes were mixed with 1:1 M ratio (+) or 10:1 M ratio (++) of non-tagged PDI or PDI3FLAG variant followed by incubation at RT and purification by IMAC. Control samples of Ero1 complexes without external PDI variant and PDI variants without Ero1 complex were treated similarly and analyzed additionally before IMAC. R, reduced; NEM+NR, NEM-treated nonreduced; and M, molecular weight marker.

(Cys131 in Ero1α) plus two possible regulatory disulfides (Cys208-Cys241 and Cys85-Cys391) (Appenzeller-Herzog et al, 2008; Baker et al, 2008; Ramming et al, 2015), Ero1α activity has been shown to be tightly controlled, whereas Ero1β is suggested to be loosely regulated reminiscent of the deregulated Ero1α C104/131A mutant (Appenzeller-Herzog et al, 2008; Wang et al, 2011). This looser regulation leads to augmented ER oxidation when overexpressed in mammalian cells (Appenzeller-Herzog et al, 2008).

Disulfide formation in vivo has been shown to be very rapid (Appenzeller-Herzog et al, 2010). Previous in vitro kinetic studies based on oxygen consumption assays using human Ero1 with intact regulatory cysteines toward the physiological substrate, PDI, have reported a maximum turnover of ~0.25 s$^{-1}$ for Ero1α (Araki & Nagata, 2011) and 0.32 s$^{-1}$ for Ero1β (Wang et al, 2011), which are inconsistent

with the fast in vivo oxidation rates (as discussed in Appenzeller-Herzog et al [2010]). Detailed kinetic studies on human Ero1 systems, including characterization of the mentioned regulatory differences between Ero1α and Ero1β, have not been reported yet.

In this study, we demonstrate that the complete time course of an Ero1 oxygen consumption trace including the initial activation lag phase can be modeled by a novel nonlinear regression method yielding detailed kinetic parameters. We report differences in activation mechanisms between Ero1α and Ero1β and novel regulatory mechanisms for both Ero1 isoforms, including a stable mixed disulfide complex formed between Ero1 and PDI, as well as high affinity and cooperativity of binding for oxygen. These findings give novel insights into how Ero1 functions and provide the first biochemical evidence for the mechanisms of how cells trade the need

to maintain disulfide bond formation at low oxygen concentrations, while shutting it off under severe hypoxia.

# Results

### Production of wild-type Ero1α and Ero1β

Previously published data on human Ero1 enzymes have consistently shown low in vitro activity (Inaba et al, 2010; Araki & Nagata, 2011; Wang et al, 2011), possibly caused by either (i) utilization of mutant variants of Ero1 and/or (ii) the use of in vitro refolding to generate the protein leading to heterogeneously folded protein. This low activity has resulted in the focus on hyperactive mutants (e.g., C104A and C131A for Ero1α), which lack a regulatory disulfide (e.g., Masui et al, 2011). To examine the kinetics of in vivo–folded wild-type human Ero1α and Ero1β, both human Ero1 paralogs were expressed in *Escherichia coli* using a system for the production of folded disulfide-bonded proteins, which includes PDI as a catalyst of disulfide bond isomerization (Nguyen et al, 2011a; Gaciarz et al, 2016). Both human Ero1 isoforms were produced in the soluble fraction. Unexpectedly, PDI co-purified with both human Ero1 isoforms in approximately 1:1 ratio (Fig 1B, lanes 1 and 2, compare with lane 3; Tables S1 and S2), suggesting that stable heterodimeric complexes formed in vivo. A disulfide bond linked the subunits in both complexes and ~130-kD heterodimers could be visualized by N-ethylmaleimide (NEM) trapping on a nonreducing SDS–PAGE gel (Fig 1B, lanes 5 and 6). We observed a consistent ~15% NEM trapping efficiency of the purified Ero1–PDI complexes, independent of incubation time or [NEM], suggesting that either a transient disulfide linked the subunits or that a buried cysteine caused resolution of the mixed disulfide after denaturation in SDS. Both purified Ero1 isomers contained long-range disulfides as the resolved monomer fraction migrated faster on a nonreducing SDS–PAGE gel (Fig 1B), resembling the OX states previously reported by others (Benham et al, 2000; Dias-Gunasekara et al, 2005). The purified Ero1α and Ero1β complexes had on average 0.99 ± 0.07 ($n$ = 3) and 0.92 ± 0.13 ($n$ = 5) FAD molecules per Ero1–PDI complex, respectively, and had a mixed $α/β$ secondary structure similar to PDI (Fig S1). Reverse-phase HPLC (rpHPLC) analysis indicated that circa 97% of the Ero1α was in a single, oxidized, redox state, whereas for Ero1β, 78% was oxidized and 22% lacked at least one disulfide (Fig S2).

The unexpected, stable, mixed disulfide complex between Ero1 and PDI was then characterized. As crystallization of purified heterodimers repeatedly failed, we turned our focus to examining potential cysteine residues by mutagenesis. Because PDI forms transient mixed disulfides with cysteines in the outer active site of Ero1 as part of the catalytic cycle (Mezghrani et al, 2001; Appenzeller-Herzog et al, 2008) and may reduce regulatory disulfides that also form with these cysteines (Appenzeller-Herzog et al, 2008; Inaba et al, 2010), we first introduced Cys to Ala mutations into the four cysteines comprising the outer active site and regulatory cysteines. Specifically, we generated the Ero1α C94A/C99A active site mutant, the C104A/C131A hyperactive mutant, the C99A/C104A inactivating regulatory mutant (Baker et al, 2008; Inaba et al, 2010), and the C94A/C99A/C104A/C131A cluster mutant. Corresponding

Ero1β mutants were also produced and analyzed. Unexpectedly, the co-expressed PDI co-purified with all of the expressed Ero1 mutants (Fig 1C), suggesting that the observed mixed disulfide complexes formed for reasons other than transferring oxidizing equivalents from the outer active site of Ero1 to PDI or reducing the reported regulatory disulfides.

Previously, in a disulfide-mapping experiment by mass spectrometry, Cys166 of Ero1α has been suggested to form a disulfide bond to a yet unidentified partner (Appenzeller-Herzog et al, 2008). This cysteine has received little attention and is often mutated to alanine in recombinant Ero1 production (Inaba et al, 2010; Araki & Nagata, 2011; Masui et al, 2011; Araki et al, 2013). In contrast to the results obtained with C94/C99/C104/C131A mutants, mutation of Cys166 of Ero1α and the equivalent Cys165 in Ero1β resulted in the complete loss of PDI co-purification (Fig 1D). Whereas Ero1α C166A could be isolated as a stable monomer for further studies, Ero1β C165A was found to be unstable. These results suggest a novel role for Cys166 and Cys165 of human Ero1 in interacting with PDI. This interaction seems to be critical for the production of stable wild-type Ero1β.

Having established the responsible cysteine in Ero1, we next applied active site mutagenesis to PDI to identify the relevant cysteine in this subunit. An additional PDI, ERp57, was co-expressed as alterations in the active sites of PDI may lead to incorrectly folded Ero1 because of decreased isomerase activity of the mutant PDI. Importantly, ERp57 can drive C166A Ero1α to a similar redox state as PDI can and wild-type PDI co-purifies with both Ero1 when Erp57 is co-expressed (Fig 1E), but ERp57 does not co-purify with either Ero1 in the wild-type background (Fig S3). If wild-type PDI was replaced with a mutant having all active site cysteines mutated, complex formation was abolished for Ero1α or very low yields of Ero1β were produced (Fig 1E). Using PDI a or a' domain substrate-trapping mutants, C56A or C400A, respectively, we observed formation of a wild type–like complex for both Ero1 only with PDI C400A (Fig 1E, upper panel). A stable faster migrating non-native complex was formed between PDI C56A and Ero1α but not with Ero1β. For both Ero1, the C400A PDI mutant significantly increased the NEM trapping efficiency of the mixed disulfide complex, indicating that C400 of PDI results in the loss of the intermolecular disulfide bond in the complex after conformational change (e.g., treatment with SDS). The trapping efficiency of both of the Ero1–PDI C400A mixed disulfides were found to be approximately 90%. Taken together, our data clearly demonstrate the existence of a stable complex between Ero1 and PDI mediated by a disulfide bond between Cys166 of Ero1α or Cys165 of Ero1β and Cys397 in the a' domain active site of PDI.

### Analysis of the Ero1–PDI complex

We further characterized the Ero1–PDI heterodimers by investigating the role of PDI in the complex. First, we determined whether the complex was permanent or transient by examining if the PDI molecule in the complex can exchange with an exogenous molecule of PDI. To allow observation of PDI exchange in a gel-based assay, we prepared a PDI variant with three consecutive FLAG tag sequences inserted before the ER retention signal (PDI3FLAG). His-tagged wild-type Ero1–PDI heterodimer was incubated with

different ratios of non-tagged PDI and PDI3FLAG. After incubation and purification by immobilized metal affinity chromatography (IMAC), the resulting complexes were visualized by reducing SDS–PAGE. Complexes in the control samples without added PDI remained stable during the 1 h assay and showed consistent ~1:1 co-purification of PDI (Fig 1F, compare lanes 5 and 6 with 1 and 2). When PDI3FLAG was mixed with the Ero1 complexes in a 1:1 M ratio, a clear decrease in the amount of PDI in the complex was observed with PDI3FLAG now co-purifying in ~1:1 ratio to the wild-type PDI. The results were not caused by background binding (Fig 1F, lanes 7 and 8) and the total ratio of PDI to Ero1 did not change significantly. Similar results were obtained if the redox state of both components were fixed by GSSG before mixing. These data suggest that oxidized PDI molecules were exchanging in the complex. To confirm these observations, we mixed 10-fold molar ratio of PDI3FLAG with the Ero1–PDI heterodimers. As expected for dynamic systems, the density of wild-type PDI in the complexes responded accordingly by decreasing to approximately 10% of the original PDI levels (Fig 1F, lanes 11 and 12). With PDI3FLAG in such excess, we now observed clearly elevated total PDI levels. This increased co-purification was probably caused by a secondary PDI-binding site—most likely the site used during the catalytic cycle. A comparable decrease in the equilibrated levels of PDI3FLAG were observed if 10-fold excess of wild-type PDI was added to a reaction containing Ero1–PDI complex and PDI3FLAG in 1:1 M ratio (Fig 1F, lanes 13 and 14). To summarize, we observed a precise response in the ratio of different PDI species in both Ero1 complexes after modulating external PDI levels. This demonstrates that PDI is a dynamic partner in the complex and that the mixed disulfide link formed with Ero1 via Cys166 or Cys165 can exchange with another PDI molecule.

Next, we studied the specificity of complex formation by investigating if PDI in the complex could exchange with the human PDI family members ERp46, ERp57, ERp72, P5, and PDIp. In contrast to the previous exchange assay with PDI, when the Ero1–PDI complexes were mixed with other PDI family members, no change was observed in the ratio of PDI to Ero1 compared with control samples (Fig S4). Small amounts (5–20% of Ero1 levels) of PDIp, ERp72, P5, and ERp46 were observed to co-purify with both Ero1 complexes (ERp46 co-migrating with Ero1$\beta$), most likely representing substrate binding to the active site of Ero1, as observed in the PDI exchange. Taken together, these results indicate that the Ero1–PDI heterodimers are dynamic, but complex formation is specific to PDI only.

## Enzyme kinetics: validating the system

Enzyme kinetic studies of human Ero1$\alpha$ and Ero1$\beta$ have so far been mainly limited to simple cross-comparison of oxygen consumption traces. Furthermore, enzyme activity in these assays is often assessed with mutant forms (C166A and hyperactivating) and using artificial substrates such as DTT or thioredoxin as the activity toward physiological substrates (for example, human PDI) is very low. In addition, there are no reports characterizing the lag phase that is evident in all human Ero1 oxygen consumption traces, which contains information about rearrangement of regulatory disulfides leading to activation. Using our in vivo–folded wild-type Ero1, we were, therefore, interested to carry out comprehensive kinetic

studies. To this end, we initially collected replicated oxygen consumption data sets for the Ero1–PDI complexes aiming to generate a regression method for modeling the complete oxygen consumption trace. We chose the previously described physiologically relevant coupled substrate system of PDI and GSH as targets of disulfide transfer from Ero1 (Baker et al, 2008). Clear lag phases for oxygen consumption were observed for both complexes followed by a linear period of maximal rate ($V_{max}$) and a final rapid decline of activity at low oxygen concentrations (Fig 2A). The non-catalyzed reaction showed very low activity.

Having demonstrated that our Ero1 complexes were active with three distinct phases (activation, linear limiting velocity phase, and inactivation at low oxygen concentration), we next established and validated a nonlinear regression method that could model the complete time course of oxygen consumption. Rather than directly using the output from the oxygen electrode as a function of time, we

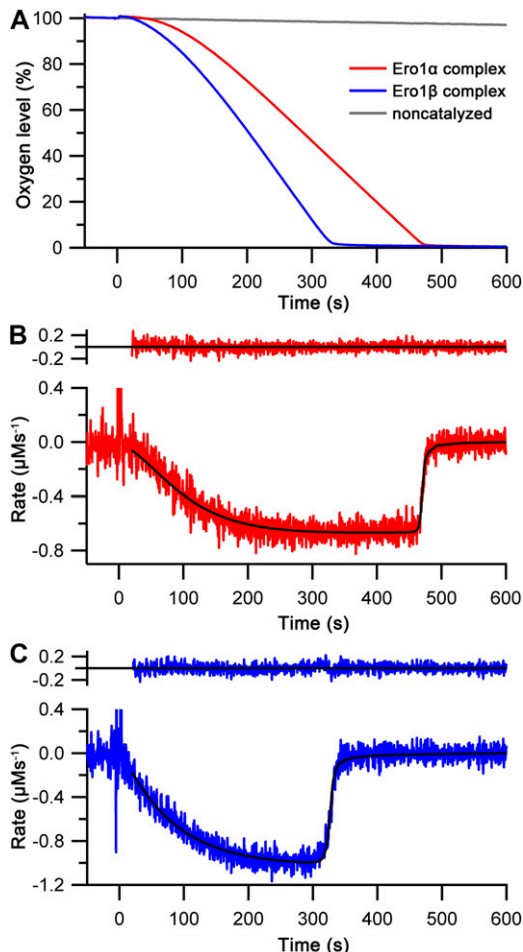

**Figure 2. Oxygen consumption kinetics of wild-type Ero1 complexes.**
**(A)** Oxygen consumption traces were collected for wild-type Ero1$\alpha$ complex or Ero1$\beta$ complex (or no enzyme) by injecting Ero1 to a substrate solution containing 10 $\mu$M PDI and 10 mM GSH. **(B)** Differentiated oxygen consumption trace for Ero1$\alpha$ complex from (A). The black curve represents the best fit combining a two-step activation process with Michaelis–Menten kinetics and a Hill coefficient for cooperativity of oxygen binding. Upper panel: Residuals for the fit. **(C)** As (B) but for Ero1$\beta$ complex that collapsed to a single step activation process at the tested [PDI].

calculated and used the derivative of this, that is, the rate of oxygen consumption at any given time point and known oxygen concentration. We hypothesized that this would then allow us to fit directly to Michaelis–Menten enzyme kinetics (rate versus oxygen concentration), modified for the activation process. With one additional factor, this hypothesis proved to be correct. The simplest model that gave random residuals to fitted data (Fig 2B and C) and integrated back to the original oxygen concentration traces (Fig S5) was based on the rate of oxygen consumption as a function of [O$_2$] and time, combined Michaelis–Menten kinetics with either a one- or two-step activation process, and included a Hill coefficient for cooperativity of oxygen binding. A similar model with no cooperativity of oxygen binding, that is, a Hill coefficient of 1, resulted in nonrandom residuals at low oxygen concentration (Fig S6). Based on these initial experiments, we were able to obtain activation rate constants, $k_{cat}$ and $K_M$, and Hill constants for oxygen binding for both complexes. Under the conditions tested with 10 $\mu$M PDI and 10 mM GSH, the Ero1$\alpha$ complex activated in a two-step process, whereas the Ero1$\beta$ complex activated faster and fitted to a single activation step (Table 1). Both enzymes showed a high $k_{cat}$ and a high affinity for oxygen (Table 1). Interestingly, the fits derived Hill coefficients >3, implying cooperativity of oxygen binding. To exclude the possibility that we had errors in our model, we tested another flavin-dependent sulfhydryl oxidase, yeast Erv1p, in similar experimental conditions fitting to the same regression model. Erv1p has not been reported to have regulatory disulfides and is thought in vivo to preferentially pass electrons to the cytochrome c system rather than to use molecular oxygen (Bihlmaier et al, 2007; Dabir et al, 2007). Consistent with this, Erv1p had no clear lag phase (Fig S7A) and our regression model fit differentiated data with random residuals (Fig S7B), revealing a low affinity for oxygen and a Hill coefficient of 1 (Table 1). These data strongly suggest that the observed high affinities for oxygen and Hill coefficients >3 for oxygen were genuine features of the human Ero1 systems and did not result from a modeling error.

### PDI-family member specificity

The assay being used to examine Ero1 activity used a couple in which the resulting Ero1-oxidized PDI-family member was reduced by GSH. To verify that the [GSH] that is typically used in Ero1 activity assays was not limiting our kinetic measurements [GSH] was varied keeping the concentration of other components constant. As expected, the $V_{max}$ of the Ero1$\beta$ complex had reached the maximal rate by 10 mM GSH (Fig S8A). Therefore, 10 mM GSH was used in all subsequent assays. Next, the PDI concentration was varied. We

observed a rapid increase in rate with increasing concentration of PDI with a best fit maximal $k_{cat}$ of 1.06 ± 0.02 s$^{-1}$ and $K_M$ of 2.8 ± 0.3 $\mu$M (Fig 3A). The Hill coefficients (3.2 ± 0.7) obtained at all [PDI] were consistent with the initial experiment. Next, we carried out a similar PDI titration assay for the Ero1$\alpha$ complex (Fig S9). Compared with the Ero1$\beta$ complex, these results showed significantly lower affinity for PDI, $K_M$ of 8.5 ± 1.1 $\mu$M, but similar $k_{cat}$ of 0.78 ± 0.03 s$^{-1}$. The Hill coefficient (4.0 ± 1.2) showed consistency with the initial results at all [PDI]. The $K_M$ for PDI of Ero1$\alpha$ is of the same order of magnitude as the $K_D$ (2.1 $\mu$M) reported for Ero1$\alpha$ C166A using surface plasmon resonance (Inaba et al, 2010). These data suggest that although not limiting Ero1$\beta$ kinetics, 10 $\mu$M PDI is a subsaturating concentration for Ero1$\alpha$. However, as it would have been impractical to carry out further experiments at saturated PDI concentrations (>30 $\mu$M), we decided to continue using 10 $\mu$M external PDI for both Ero1 complexes in all subsequent experiments.

As both Ero1 complexes transferred disulfides efficiently to PDI, we next investigated their ability to use other PDI family members as substrates. Oxygen consumption data were collected for both complexes with or without added PDI, PDIp, ERp57, ERp46, ERp72, or P5. All oxygen consumption rates, except for ERp72, fit consistently to the model with one- or two-step activation and a Hill coefficient >3. Reactions using ERp72 as a substrate showed a clear time-dependent loss of activity (Fig S10) with the best fit being achieved by the addition of an inactivation step. Both complexes without the addition of an external PDI family member showed activity (Fig 3B and C). The Ero1$\alpha$ complex showed low activity with a $k_{cat}$ of 0.08 ± 0.02 s$^{-1}$, whereas the Ero1$\beta$ complex had a $k_{cat}$ of 0.22 ± 0.01 s$^{-1}$. This difference may arise because of the fact that circa 22% of the Ero1$\beta$ used lacked at least one disulfide (Fig S2B), potentially the regulatory disulfide reduced by PDI. Consistent with previous assays, the addition of PDI accelerated activity significantly, 7.1-fold for the Ero1$\alpha$ complex and 4.8-fold for Ero1$\beta$ complex. Other PDI family members increased activity either marginally or not at all, with the exception of PDIp that increased the rate of reaction of the Ero1$\beta$ complex 3.7-fold, almost to the level of PDI. These data demonstrate, consistent with previous studies (Araki et al, 2013), that other PDI family members are generally poor substrates in vitro for both human Ero1 isoforms. PDIp shows exception to this rule by accepting disulfides efficiently from Ero1$\beta$ but not from Ero1$\alpha$.

### Ero1 activation

Ero1 activity proceeds via a lag phase not observed for Erv1p (Fig S7), consistent with regulatory disulfides being rearranged during

**Table 1. Kinetic parameters for Ero1 complexes and Erv1p.**

| Enzyme (1 $\mu$M) | Activation rate constants | | Halftime of activation (min$^{-1}$) | $k_{cat}$ (s$^{-1}$) | $K_M$ ($\mu$M) | Hill constant |
| --- | --- | --- | --- | --- | --- | --- |
| | $k_1$ (min$^{-1}$) | $k_2$ (min$^{-1}$) | | | | |
| Ero1$\alpha$–PDI | 1.04 ± 0.04 | 1.12 ± 0.06 | 1.56 ± 0.08 | 0.64 ± 0.04 | 5.0 ± 0.8 | 5.2 ± 0.7 |
| Ero1$\beta$–PDI | 1.11 ± 0.03 | n.m. | 1.14 ± 0.05 | 1.00 ± 0.02 | 7.6 ± 1.2 | 3.1 ± 0.5 |
| Erv1p | n.m. | n.m. | n.m. | 0.41 ± 0.03 | 89 ± 18 | 1.0 ± 0.1 |

n.m., not measurable.

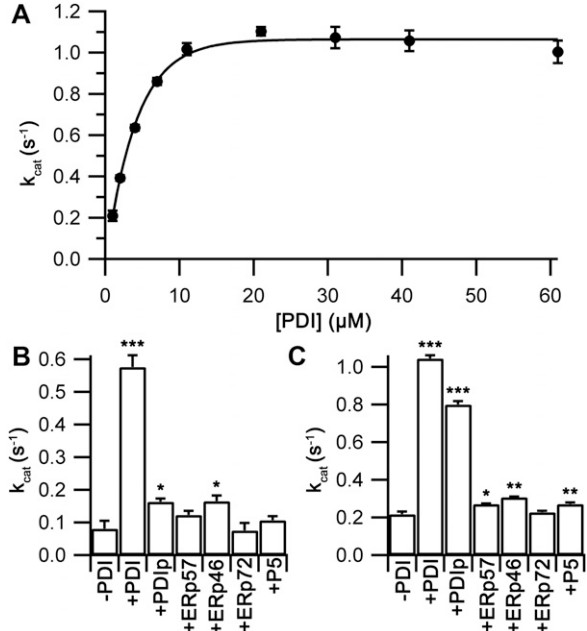

**Figure 3. Substrate affinity and specificity for wild-type Ero1 complexes.**
**(A)** A plot of $k_{cat}$ values, obtained as in Fig 2 with varying [PDI], versus total [PDI] for Ero1$\beta$ complex with an exponential fit ($n$ = 3; mean ± SD). **(B, C)** $k_{cat}$ values ($n$ = 2 for ERp72, $n$ = 3 for others; mean ± SD) for Ero1$\alpha$ complex (B) and Ero1$\beta$ complex (C) in the presence or absence of 10 $\mu$M external PDI family member. Significance levels: $*P < 0.05$, $**P < 0.01$, and $***P < 0.001$; $t$ test with two-tailed distribution and two-sample unequal variance.

activation. The lag phase of the Ero1 kinetics has not received any detailed attention yet despite this phase being evident in all reported oxygen consumption traces. Hence, we were interested what information we could extract from our experimental data about the regulatory processes leading to activation. We first extracted activation kinetics from the GSH and PDI titration experiments. Activation of the Ero1$\beta$ complex was not affected by GSH concentration (Fig S8B), indicating that the activation steps observed did not reflect reduction of regulatory disulfides by GSH. In contrast, when activation rate constants from both PDI titration experiments were plotted, a strong [PDI] dependence was observed. For the Ero1$\beta$ complex, one rate constant was observed that was independent of [PDI] and one rate constant that was dependent on [PDI] and was too fast to measure at [PDI] >6 $\mu$M. Combining these gave rise to a halftime for activation that showed a midpoint for [PDI] of 4.6 ± 0.8 $\mu$M (Fig 4A), comparable with the $K_M$ of the complex for PDI as a substrate (Fig 3A). In contrast, the Ero1$\alpha$ complex showed a two-step activation process with both activation steps showing a dependence on [PDI] (Fig 4B). For both activation steps, the midpoint for PDI dependence was of the same order of magnitude as the $K_M$ for PDI as a substrate (13.3 ± 2.1 $\mu$M and 19.0 ± 3.3 $\mu$M compared with a $K_M$ of 8.5 ± 1.1 $\mu$M).

Because PDI was a potent activator of both complexes, we were interested if other PDI family members showed similar enhancement of activation. Interestingly, PDIp and ERp46 activated the Ero1$\alpha$ complex as rapidly as PDI (Fig 4C) despite having a low $k_{cat}$ (Fig 3B) and PDIp also activated the Ero1$\beta$ complex (Fig 4D). Other tested combinations did not activate Ero1 complexes faster than the negative controls.

Together, these data demonstrate clear differences in activation steps between the human Ero1 complexes. Consistent with previous results (Zhang et al, 2014), these regulatory steps can be modulated by different PDI family members, but when combined with the $k_{cat}$ results (Fig 3), both complexes show a clear preference for interacting with PDI (and PDIp for Ero1$\beta$).

To try to dissect which disulfides account for the activation processes of the Ero1 systems, we prepared a series of Ero1 Cys to Ala mutants affecting either the proposed regulatory sites (Appenzeller-Herzog et al, 2008; Baker et al, 2008; Ramming et al, 2015; Kanemura et al, 2016) or our novel mixed disulfide link. Hyperactivating mutants of Ero1$\alpha$ and Ero1$\beta$ purified as stable 1:1 complexes with PDI and the complex-breaking Ero1$\alpha$ C166A purified as a monomer. Ero1$\alpha$ C208/241A, containing a proposed regulatory site at the distal side of the enzyme (Ramming et al, 2015), was produced in very low yields and had low activity, indicating structural instability. The corresponding Ero1$\beta$ C207/240A was unstable and could not be analyzed. Oxygen consumption data were collected similarly to previous assays keeping total [PDI] constant. For Ero1$\alpha$, the wild-type, hyperactivating mutant, and C208/241A mutant all fitted consistently to the model with two-step activation and Hill coefficient >3. In contrast, the Ero1$\alpha$ C166A mutant showed a time-dependent loss of activity during the assay and fitted best to a model with a single activation step and an inactivation rate (Fig S11). We observed no change in activation rate with the hyperactive mutant of Ero1$\alpha$, whereas both the C208/C241A and the C166A mutant showed enhanced activation rates (Fig 4E). $k_{cat}$ was increased slightly in the hyperactive mutant, consistent with literature (Inaba et al, 2010; Araki & Nagata, 2011), whereas the C166A mutant showed 67% activity and the distal regulatory site mutant only 43% activity of the wild-type complex (Fig 4F). In contrast to the Ero1$\alpha$ results, the hyperactivating Ero1$\beta$ mutation increased activation but decreased $k_{cat}$. These results further clarify the differences in activation between the two human Ero1 systems.

The two activation steps for Ero1$\alpha$ are dependent on [PDI], suggesting that PDI exchange may be involved in the activation process. To examine this, we mixed the Ero1$\alpha$ complex with 3 $\mu$M of PDI3FLAG under conditions used in the enzyme assay, extracted samples, and quenched the reaction with NEM at set time points. Wild-type complex disappeared with time (Fig S12A), consistent with a two-step activation process and a halftime of ~1.5 min (Fig S12B). Concomitant with this, a higher molecular weight complex in which the PDI in the complex had been exchanged with PDI3FLAG appeared on the gel (Fig S12C), reaching an apparent plateau after 300 s and with a halftime of ~1.7 min, consistent with activation being linked to PDI exchange.

## Discussion

Detailed mechanistic characterization of wild-type human Ero1 produced using a bacterial expression system allowed the elucidation of kinetic parameters and the identification of novel regulatory mechanisms. Specifically, we identified (i) the formation of a stable mixed disulfide complex of PDI and Ero1 in the inactive state, (ii) a high Hill coefficient for oxygen, and (iii) differences in the

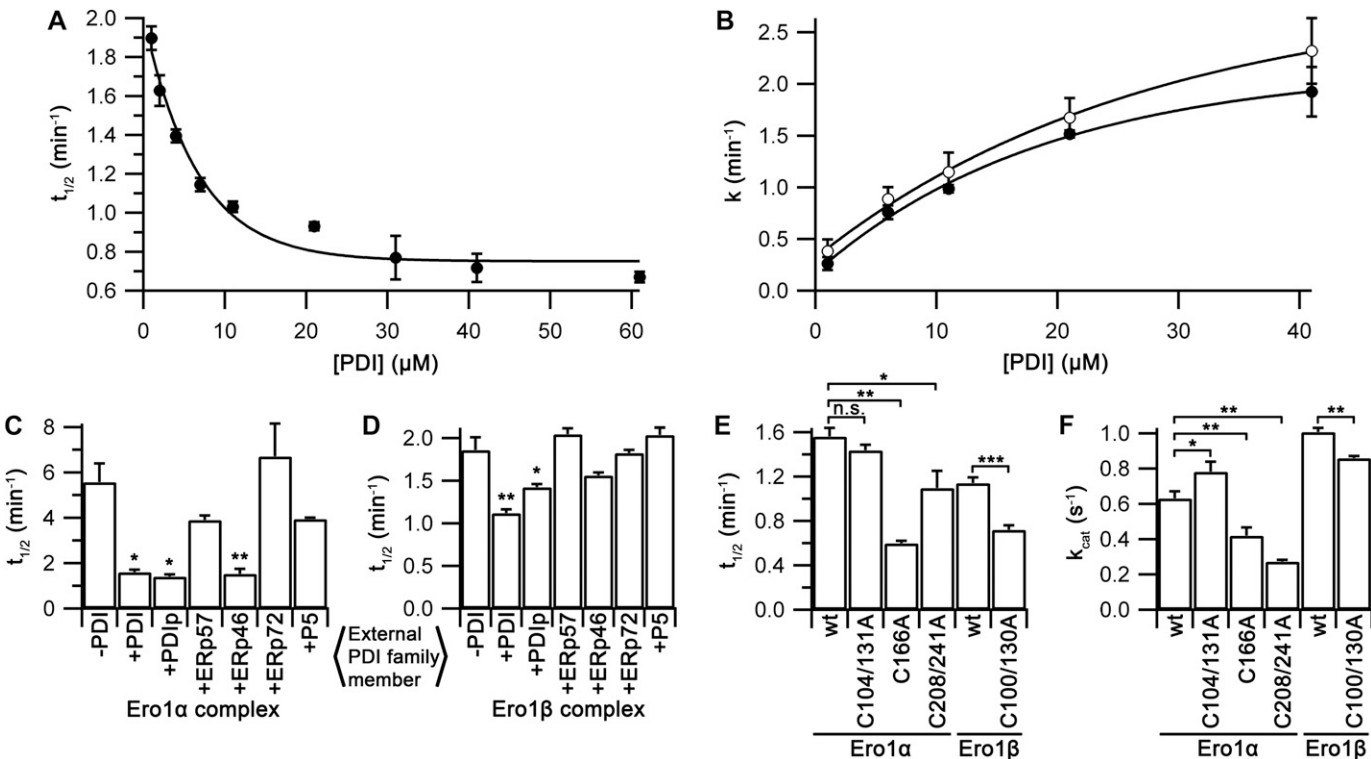

**Figure 4. Activation kinetics of Ero1α and Ero1β complexes.**
**(A, B)** [PDI] dependence of halftime of activation of Ero1β complex (A; $n = 3$, mean ± SD) and individual activation rate constants of Ero1α complex (B; $n = 2$, mean ± SD). **(C, D)** Halftime of activation of Ero1α complex (C) and Ero1β complex (D) for different PDI family members ($n = 2$ for ERp72 and $n = 3$ for others; mean ± SD). **(E, F)** Halftimes of activation (E) and $k_{cat}$ values (F) for complexes of wild-type Ero1α and Ero1β and their hyperactivating mutants C104/131A and C100/130A as well as monomeric Ero1α C166A and C208/241A mutants ($n = 3$; mean ± SD). Significance levels: *$P < 0.05$, **$P < 0.01$, ***$P < 0.001$, and n.s., not significant; $t$ test with two-tailed distribution and two-sample unequal variance.

rate-limiting steps of activation of Ero1α versus Ero1β. All of these have significant implications for the in vivo mechanisms and regulation of disulfide bond formation in the ER.

Overall, human Ero1α and Ero1β share a number of similarities. They both form a mixed disulfide linked complex with PDI in the inactive state, preferentially use PDI over other PDI-family members, have a similar $K_M$ and Hill coefficient >3 for oxygen, and have two measureable activation steps with a minimal activation halftime of 59 and 45 s for Ero1α and Ero1β, respectively, under the conditions tested.

The formation of the novel, stable, mixed disulfide complex between Ero1 and PDI highlights the importance of Cys166 (Ero1α) and Cys165 (Ero1β), which have previously been stated non-functional (Araki & Nagata, 2011). Although the OX2 state of Ero1 is currently regarded as the inactive state, incapable of transferring oxidizing equivalents to PDI (Appenzeller-Herzog et al, 2008; Baker et al, 2008), our results suggest that the use of C166A mutants and the inability to efficiently trap the mixed disulfide complex reported here may have obscured the more complex reality. This regulatory complex has probably been previously observed in vivo as a mixed disulfide was observed to form concomitantly with the appearance of the OX2 state of Ero1α after translation into intact and functional ER of semi-permeabilized cells (Benham et al, 2000), but this has not been previously followed up. The formation of the complex probably results in the regulatory disulfides being inaccessible and,

hence, has a significant impact on activation. Of the 404 sequences from Ero1 family members from chordates in InterPro that are sufficiently complete to cover this region, only one lacks a cysteine at the equivalent position to Cys166/C165 (A0A2K6D959, Ero1β from *Macaca nemestrina*), implying the formation of this complex is conserved.

In addition to the novel regulatory complex, we also characterized a second novel modulator of Ero1 activity. Specifically, molecular oxygen showed not only high substrate affinity but also a Hill coefficient >3. This has important implications for regulation of oxidative folding in vivo. By having a high Hill coefficient and high affinity for oxygen, Ero1 will retain high activity, even at low oxygen concentrations. For example, Ero1α ($K_M = 5.0$ μM and Hill coefficient = 5.2) retains 90% of its activity at 7.6 μM oxygen (Fig 5A). Equally importantly, under hyper-hypoxic conditions, the activity of Ero1α will be rapidly reduced such that Ero1α will have only 10% activity at 3.3 μM oxygen. This drastic reduction in activity over an oxygen concentration range of only 4.3 μM is consistent with the cell requiring maximal rates of oxidative folding even under hypoxic conditions but wanting to conserve oxygen for other more essential metabolic processes under hyper-hypoxic conditions. The essentiality of a high Hill coefficient in this process can be seen from modeling the effects with a similar $K_M$ but Hill coefficient = 1 (Fig 5A). Under these conditions, 10% of activity would be seen at 0.56 μM oxygen and 90% at 45.0 μM, a range of 44.4 μM and more than

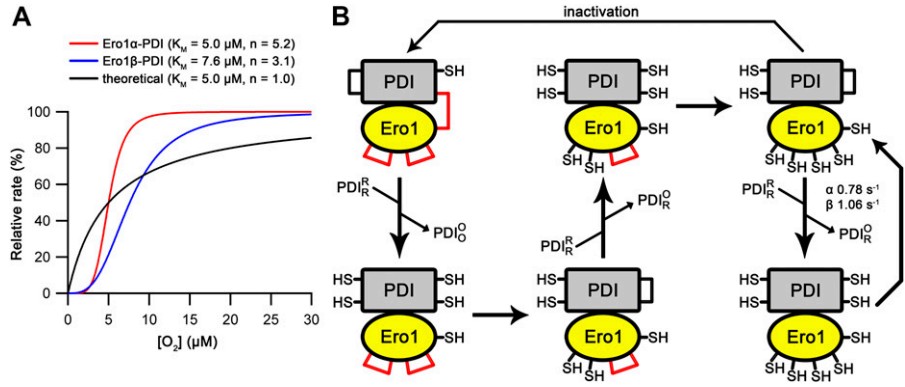

**Figure 5. Effect of Hill constant on Ero1 kinetics and model for Ero1 activation, catalytic cycle, and inactivation.**
**(A)** Comparison of relative rates of fully activated Ero1 complexes using experimentally derived values for $K_M$ and Hill coefficient to a theoretical case with $K_M$ equivalent to that of the alpha complex and Hill coefficient of 1. **(B)** Model for the catalytic cycle of Ero1. The inactive state of Ero1 (upper left corner) contains three regulatory disulfides, including the mixed disulfide between Ero1α C166 or Ero1β C165 and PDI. Straight vertical lines represent PDI exchange steps in which (semi)oxidized PDI is released with incoming reduced PDI. The first two PDI exchange steps are slower than that during the catalytic cycle because of the presence of Ero1 regulatory disulfides. Straight horizontal lines represent regulatory disulfide reduction steps. After the three regulatory disulfides have been reduced, Ero1 enters the fast catalytic cycle with indicated turnover numbers. The fully active Ero1 bound to semi-oxidized PDI state has a kinetic partitioning event. If reduced PDI is predominant, the catalytic cycle continues. Once the levels of reduced PDI decrease, partitioning to inactivation increases. The steps of inactivation include mixed disulfide formation and auto-oxidation of Ero1 regulatory disulfides, but the order is unclear.

10 times that observed for modulation of Ero1α activity. The required retention of high activity under hypoxic conditions but sharp reduction below a critical threshold is only possible with a high Hill coefficient and suggests that Ero1 is fine-tuned to use oxygen as a terminal acceptor in a manner that is non-deleterious to the cell, that is, that does not result in cell death under hyper-hypoxic conditions. The mechanisms by which cooperativity of oxygen binding occurs remain to be elucidated. There is no evidence that the Ero1–PDI complex forms higher molecular weight complexes, for example, trimers of Ero1–PDI dimers, during its catalytic cycle so the mechanism of cooperativity must be intra- rather than intermolecular. Hence, either multiple oxygen-binding sites exist within Ero1 or it has allosteric-like activation mechanisms similar to glucokinase, which is monomeric and has only one substrate-binding site. Cooperativity of substrate binding in glucokinase arises from order–disorder transitions in the active site (Larion et al, 2012; Whittington et al, 2015).

Although Ero1α and Ero1β share many similar properties, they differ significantly in their activation, and this may explain why two isoforms are needed. All reported oxygen consumption traces for human Ero1 contain a lag phase. This translates as the necessity to undertake activation step(s), including the rearrangement of regulatory disulfides (Sevier et al, 2007; Appenzeller-Herzog et al, 2008), to reach the fully active state. Using our novel nonlinear regression method, we were able to examine the kinetic details of activation of human Ero1 for the first time. Although the minimum time observed for activation is of the same order of magnitude for the human Ero1 orthologs (59 s for Ero1α versus 45 s for Ero1β), there are clear differences in the activation steps. Ero1α activated by two steps, both of which showed dependence on external PDI. In contrast, Ero1β showed one PDI-independent activation step and one PDI-dependent step, which became unmeasurably fast by 6 μM PDI. This has important consequences for in vivo function. Ero1α has a strong dependence on the availability of reduced PDI for both activation steps (Fig 4) and for catalytic turnover (Fig 3)—although PDIp and ERp46 can replace PDI for the activation steps. In contrast, Ero1β has less dependence on the availability of reduced PDI for one activation step and activates by a PDI-independent process for the other step. Hence, Ero1β will show significantly greater activity

under conditions of low abundance of reduced PDI, whereas Ero1α activity has a strong dependence on the availability of this substrate.

From our data, we have developed a scheme that describes the activation events that lead to the catalytic cycle and back to inactivated state (Fig 5B). The oxidized Cys166/165-Cys397 Ero1–PDI heterodimer is the inactive state of the enzyme and this heterodimer is stable in the absence of exogenous PDI allowing purification of the complex. In the presence of exogenous PDI, PDI exchange occurs (Figs 1F and S12). If exogenous oxidized PDI exchanges, there is no net change in the complex (Fig S13). In contrast, if exogenous reduced PDI exchanges, Ero1 proceeds to activation (Figs 5B and S13). The PDI family members ERp57, ERp72, and P5 are unable to exchange with PDI in the complex and, hence, result in slow activation rates even when present in molar excess (Fig 4). This demonstrates the specificity for the exchange beyond having a WCGHC active site motif and suggests that something is required in addition to thiol-disulfide exchange involving Cys166/165. This PDI exchange combined with (i) the PDI dependence of both activation steps of Ero1α and (ii) the $K_M$ values for substrate PDI and for PDI dependence of activation being similar (8.5 ± 1.1 μM for catalysis cf. 13.3 ± 2.1 μM and 19.0 ± 3.3 μM for activation), suggests that activation proceeds via exchange of exogenous reduced PDI with PDI from the complex and with both the incoming and outgoing PDI competing for the same site in Ero1, that is, the β hairpin (Masui et al, 2011). Each PDI exchange would result in the reduction of one (or more) disulfide bonds, resulting in the generation of the active state of Ero1. PDI exchange would then continue as part of the catalytic cycle, with a turnover of 0.64 s$^{-1}$ for Ero1α. The fast catalytic turnover with respect to the kinetics of activation suggests that (partial) inactivation does not form part of the catalytic cycle. Instead, if there is no reduced PDI available to exchange with the Ero1-associated oxidized PDI, the regulatory disulfides are formed, including the mixed disulfide with PDI. Because oxidized PDI has been demonstrated to associate with Ero1α with a similar rate to reduced PDI (Masui et al, 2011), regulatory disulfides may form via cycles of exchange of oxidized PDI. However, a simpler model would use auto-oxidation of Ero1 to form the regulatory disulfides (Zhang et al, 2014). In this scheme, the same molecule of oxidized PDI would

interact with Ero1 during the whole pathway for inactivation, keeping the regulatory disulfides buried and inaccessible for further reduction until the oxidized PDI was exchanged with a reduced molecule of PDI.

Although Ero1β has a different PDI-(in)dependence for activation, this does not necessarily mean a different reaction pathway is required. Instead, the simplest explanation for the PDI-(in)dependence of Ero1β activation is that the PDI exchange reactions (which are limiting for Ero1α) are faster for Ero1β, such that the disulfide rearrangement reactions become rate limiting. As such, the overall scheme is identical as that for Ero1α (Fig 5B), but the relative rates of individual steps differ.

What are the regulatory disulfides being reduced during activation? Our scheme contains three regulatory disulfides that need to be reduced. (i) Reduction of the characterized regulatory switch, Cys94-Cys131 in Ero1α (Appenzeller-Herzog et al, 2008; Baker et al, 2008) and Cys90-Cys130 in Ero1β (Wang et al, 2011). This step is very fast for both mutants resulting in only a slight difference in halftime of activation for the hyperactive mutant (Fig 4E). Instead the increase in in vivo activity observed (Appenzeller-Herzog et al, 2010) may result from increased $k_{cat}$ (Fig 4F). (ii) Reduction of the novel mixed disulfide between Ero1 and PDI during the first PDI exchange step. Mechanistically, this most likely involves nucleophilic attack by the C-terminal active site cysteine (Cys400) on the mixed disulfide. Consistent with this, the C166A mutation in Ero1α C166A both activates faster (Fig 4E) and shows only one activation rate constant. (iii) The requirement for an additional PDI exchange step strongly suggests that another disulfide bond is reduced within Ero1. Because the C208A/C241A tunnel disulfide mutation showed a two-step activation process, this is most likely not the last regulatory disulfide, despite being suggested to modulate tunneling of oxygen to FAD (Ramming et al, 2015). The only remaining candidate for the final reductive activation step would be reduction of the long-range disulfide Cys85-Cys391 (or Cys81-Cys390 in Ero1β) situated near the bound FAD and Cys94-Cys131 disulfide (Inaba et al, 2010). This disulfide was initially thought to be regulatory (Baker et al, 2008) but later was suggested to be structural instead (Araki & Nagata, 2011; Zhang et al, 2014). Unfortunately, we were unable to produce this mutant for activation studies to confirm this hypothesis, as also reported by others (Baker et al, 2008; Chambers et al, 2010). Once activated, Ero1 proceeds to a catalytic cycle in which it passes oxidizing equivalents from the inner active site to the outer active site to the a' domain of PDI (Baker et al, 2008; Wang et al, 2009).

# Materials and Methods

## Molecular biology

Expression vectors (Table S3) were prepared using standard molecular techniques, including site-directed mutagenesis with the QuickChange site-directed mutagenesis kit (Stratagene).

Mature human Ero1α (encoding residues E24-H468) and Ero1β (Q34-R467, natural D129V, and H465Q variant) were cloned previously (Nguyen et al, 2011b). From these vectors, they were subcloned into a modified pET23 vector that includes an N-terminal MHHHHHHM-tag (His tag or $H_6$) and has an additional SpeI cloning site between EcoRI and SacI sites (Nguyen et al, 2011a). From here, they were subcloned by XbaI/XhoI digestion to the SpeI/SalI site of a modified pMAL-c2x vector that does not encode for maltose binding protein fusion (Hatahet & Ruddock, 2013). The lost XhoI site of the resulting plasmids was restored by mutagenesis. Genes for codon-optimized mature human ERp57 (encoding resides S25-L505) and mature human P5 (encoding residues L20-L440) were obtained from GenScript. A polycistronic Ero1-ERp57 vector was created by cloning ERp57 to the restored SpeI/XhoI site in the Ero1 vector.

Expression vectors for the mature forms of the human PDI family members PDI, PDIp, ERp72, and ERp46 with an N-terminal His tag in the pET23 background have been described (Alanen et al, 2003). Similar vectors but with a tac promoter in place of the T7 promoter (Hatahet et al, 2010) were made for ERp57 and P5.

Construction of the polycistronic CyDisCo plasmid pMJS205 encoding Erv1p and wild-type codon-optimized PDI has been described previously (Gaciarz et al, 2016). Similar vectors with mutant forms of PDI were similarly constructed.

A codon-optimized mature human PDI construct with a triple FLAG-tag sequence (PDI3FLAG, encoding residues DYKDDDDK-DYKDDDDKDYKDDDDK) inserted between alanine 503 and valine 504 was created by cloning a synthetic codon-optimized fragment to an internal ClaI and BamHI site. This construct along with wild-type codon-optimized mature human PDI were cloned into the NdeI/BamHI site of a pET23-based vector with a tac promoter, which included an in-frame N-terminal His-tag followed by a tobacco etch virus (TEV) protease cleavage site (sequence −MHHHHHHSSGVDLGTENLYFQSHM−).

PDI family members with a His-tagged tDsbC fusion partner were prepared by cloning the abovementioned PDI family member genes into NdeI/BamHI site of a modified pET23-based vector with a tac promoter encoding for TEV protease cleavage site (−GGGSGSENLYFQGSHM−) between the fusion and the gene (Zhang et al, in preparation).

All plasmids generated were sequenced (Biocenter Oulu core facility) to ensure that there were no errors in the cloned gene.

## Generation of BL21(DE3)Δ*trxA*Δ*trxC* strain

BL21(DE3)Δ*trxA*Δ*trxC* was generated with standard P1 phage transduction (Thomason et al, 2007). Briefly, the P1 phage was first cleaned by growing in recipient wild-type BL21(DE3) cells (Novagen). Next, P1 lysate was prepared from the Keio collection donor strain JW2566-1 (Baba et al, 2006), which is a K12 derivative carrying mutation trxC750(del)::kan and used for P1 transduction in the recipient BL21(DE3) strain. After successful transfer of the desired genetic modification, the kanamycin resistance determinant was removed via Flp-catalyzed excision using pCP20 (Cherepanov & Wackernagel, 1995). The second mutation, that is, Δ*trxA*, was generated similarly via P1 phage transduction using Keio collection strain JW5856-2 carrying mutation trxA732(del)::kan as a donor strain. The transfer of desired genetic modifications was screened and verified by colony PCR.

## Protein expression, purification, and analysis

His-tagged Ero1 variants were expressed from modified pMAL-c2x vectors with co-expression of CyDisCo components (yeast Erv1p and

human PDI) from pMJS205 (Gaciarz et al, 2016) or an equivalent vector containing PDI active site mutants, in *E. coli* strain BL21(DE3)Δ*trxA*Δ*trxC*. His-tagged PDI, PDIp, ERp57, ERp72, P5, and Erv1p were expressed in either BL21(DE3)pLysS or BL21(DE3) and His-tagged ERp46 in BL21(DE3)Δ*trxA*Δ*trxC* without CyDisCo.

Cells expressing Ero1 and Erv1p were cultivated in EnPressoB (BioSilta Oy) media as instructed by the manufacturer and PDI family members in autoinduction media (Li et al, 2011) at 30°C in either 250-ml shake flasks (10% fill volume, 250 rpm) for small-scale mutational analyses or in 500-ml shake flasks (10% fill volume, 250 rpm) and 2.5 liters Ultra Yield flasks (10–20% fill volume, 300 rpm; Thomson Instrument Company) for large-scale production. All cultures were supplemented with 100 $\mu$g/ml ampicillin with an additional 35 $\mu$g/ml chloramphenicol for any double plasmid expressions and an additional 25 $\mu$g/ml kanamycin for the knockout strain. Ultra Yield expressions were supplemented with antifoam 204 (Sigma-Aldrich). After 16 h, EnPressoB cultures were boosted as instructed by the manufacturer and induced with 0.5 mM IPTG. EnPressoB cultures were harvested by centrifugation (3,200–5,000 $g$, +4°C, 20 min) 24 h post-induction and autoinduction cultures 26–30 h after inoculation. Pellets for small-scale analyses were resuspended in a culture volume of 50 ml sodium phosphate, pH 7.3, 100 $\mu$g/ml egg white lysozyme, and 20 $\mu$g/ml DNase and frozen. Large-scale expressions were pooled and frozen as pellets.

Cells from small-scale expressions were lysed by two rounds of freeze–thawing, cleared by centrifugation (3,200 $g$, +4°C, 20 min), and purified by IMAC using HisPur Cobalt Agarose (Thermo Fisher Scientific) essentially as described (Gaciarz et al, 2016) with minor modifications. Briefly, after clearing the lysate, proteins were bound to HisPur resin using 0.5 ml bed volume, washed with 50 mM sodium phosphate, pH 7.3, 300 mM NaCl, 15 mM imidazole, and eluted with 50 mM sodium phosphate, pH 7.3, and 200 mM imidazole.

The pellets of large-scale Ero1 and Erv1p expressions were resuspended to 40% culture volume of 50 mM sodium phosphate, pH 7.3, 150 mM NaCl, and 5 mM imidazole (lysis buffer) and lysed by sonication. Cell lysates were clarified by centrifugation (30,000 $g$, +4°C, 40 min) and proteins were bound to HisPur Cobalt Agarose mixing for 30 min at +4°C. Resin was collected, washed with at least 40× bed volume of lysis buffer, cleared by 20 mM sodium phosphate, pH 7.3, and eluted with 20 mM sodium phosphate, pH 7.3, and 200 mM imidazole (pH 7.0 for Erv1p). IMAC eluates were diluted 2.5-fold with sodium phosphate, pH 7.3 (7.0 for Erv1p), and bound to a 6-ml Resource Q anion exchanger (Ero1 proteins; GE Healthcare) or a 5-ml HiTrap SP FF cation exchanger (Erv1p; GE Healthcare) pre-equilibrated with the same buffer. Ero1 proteins were eluted by increasing NaCl from 0 to 0.6 M in 20 mM sodium phosphate, pH 7.3, at a shallow rate of 1% NaCl increment per column volume (CV) for Ero1$\alpha$ peak and 5% per CV for Ero1$\beta$. Erv1p was eluted with faster gradient in 5 CV. Purity was assessed by SDS–PAGE, the purest fractions were pooled, concentrated by Amicon Ultra centrifugal filters (Millipore), and further purified by size-exclusion chromatography with a Superdex 200 16/600 HiLoad column (GE Healthcare) pre-equilibrated with 20 mM sodium phosphate, pH 7.0, and 150 mM NaCl (activity assay buffer). The purest fractions were concentrated to 25 $\mu$M, aliquoted, flash-frozen in liquid nitrogen, and stored at −70°C. Wild-type Ero1$\alpha$ and Ero1$\beta$, along with many mutants of these enzymes (see main article), co-purified as a heterodimeric complex with PDI.

Large-scale PDI family member expressions were purified by HisPur Cobalt Agarose IMAC and anion exchange essentially as described above using a gradient of 2% NaCl increment per CV for anion exchange, omitting size-exclusion chromatography step, and exchanging the buffer of the purest fractions to activity assay buffer by Amicon Ultra centrifugal filters (Millipore) before concentration to 100 $\mu$M and processing to storage at −70°C.

Purification of non-tagged wild-type PDI and PDI3FLAG for PDI exchange assays used a TEV protease–cleavable His tag, whereas purification for other PDI family members used a TEV protease–cleavable His-tagged tDsbC fusion partner (Zhang et al, in preparation). Both sets of purification involved an extra step of protease treatment before anion exchange. For this step, IMAC eluates were buffer-exchanged to 20 mM sodium phosphate, pH 7.3, 150 mM NaCl with PD-10 columns, mixed with His-tagged TEV protease for 16–18 h at +4°C, then with HisPur resin in the presence of 10 mM imidazole for 30 min at +4°, and the flowthrough was collected. Success of cleavage was confirmed by SDS–PAGE and anion exchange was executed for the flowthrough as described above.

Protein concentrations were determined using calculated molecular masses and absorption coefficients at 280 nm (Table S4; Gasteiger et al, 2005). For Ero1 and Erv1p preparations, contribution of FAD was additionally estimated and subtracted (see below).

The molecular weight of all purified proteins was confirmed by electrospray ionization mass spectrometry (Tables S1 and S2).

### Determining FAD-dependent protein concentration

Concentration of all FAD-bound enzyme preparations was determined directly from the final protein stocks using calculated molecular weights and extinction coefficients for Ero1–PDI heterodimers and Ero1 mutants (Table S4) and reported extinction coefficients of FAD (Gross et al, 2006). Concentration of bound FAD was first determined by absorbance at 454 nm ($\varepsilon$ = 12,500 M$^{-1}$ cm$^{-1}$). Absorbance of FAD at 280 nm was then calculated and subtracted from the total absorbance at 280 nm by using concentration of FAD and absorbance coefficient of free FAD at 280 nm (21,300 M$^{-1}$ cm$^{-1}$).

### SDS–PAGE and sample preparation

Reduced SDS–PAGE samples were prepared by mixing protein sample directly with SDS–PAGE loading buffer containing $\beta$-mercaptoethanol. All nonreduced protein samples from purifications were treated with 20 mM NEM for 10 min at RT before adding SDS–PAGE loading buffer and heating at +95°C for 5 min. Nonreducing samples from reactions containing 10 mM GSH and requiring rapid NEM-quenching were prepared by treating the sample with 50 mM NEM in nonreducing SDS–PAGE loading buffer with immediate heating at +95°C for 5 min.

### Circular dichroism

Circular dichroism spectra were collected (as described by Gaciarz et al [2016]) in 10 mM sodium phosphate, pH 7.3.

### rpHPLC

Before rpHPLC analysis, 100 $\mu$g of purified Ero1$\alpha$ complex diluted in 500 $\mu$l of 20 mM sodium phosphate, pH 7.0, was treated with or without 2 mM DTT or 5.5 M guanidinium chloride plus 2 mM DTT for 15 min at RT. Similarly, 100 $\mu$g of Ero1$\beta$ complex was treated with guanidinium chloride in the presence of either 2 mM DTT or 20 mM NEM to avoid potential disulfide rearrangement with the Ero1$\beta$-specific unpaired Cys262 (Hansen et al, 2014). The samples were injected and bound to a $\mu$PRC C2/C18 ST 4.6/100 column (Amersham Biosciences) in a mobile phase of 0.1% TFA. Proteins were eluted with a step gradient toward 95% ACN, 0.1% TFA (0–45% in 2 CV, 45–57% in 25 CV, and 57–100% in 1 CV). Elution was monitored by changes in absorbance at 220 nm and the areas of eluted peaks were calculated using Unicorn 5.0 software (GE Healthcare).

### PDI exchange assay

1 $\mu$M of His-tagged Ero1–PDI heterodimers were incubated for 1 h at RT with or without indicated concentrations of non–His-tagged wild-type PDI, PDI3FLAG, or other PDI family members in 20 mM sodium phosphate, pH 7.3, and 150 mM NaCl. Control reactions of individual components were treated similarly. After incubation, 400 $\mu$l of the samples in the presence of 10 mM imidazole were purified by HisPur Ni-NTA 0.2 ml spin columns (Thermo Fisher Scientific) as instructed by the manufacturer with minor modifications (NaCl concentration decreased to 150 mM and pH to 7.3). Briefly, the samples were mixed with the resin for 30 min at RT. Resin was washed twice with 2 CV of wash buffer and eluted three times with 1 CV of elution buffer.

### Oxygen consumption assays

Ero1 and Erv1p activity was assessed by measuring oxygen consumption with a Clark-type oxygen electrode (Oxytherm; Hansatech Instruments). The electrode was calibrated with air-saturated activity assay buffer (20 mM sodium phosphate, pH 7.0, and 150 mM NaCl) and dithionite. Substrate solution containing varying concentrations of PDI family members, varying concentrations of GSH, 0.5 U yeast glutathione reductase (Sigma-Aldrich), 1 mM NADPH, and 2 mM EDTA in air-saturated activity assay buffer were first mixed in the electrode chamber at 25°C for 240 s. Glutathione reductase was added to avoid deviations in the kinetics by accumulating GSSG (Zhang et al, 2014). Oxygen consumption was then initiated by injecting 0.5 $\mu$M (PDI titration of Ero1$\alpha$ complex) or 1 $\mu$M (all other assays) of an Ero1 variant or 1 $\mu$M Erv1p into a sealed chamber with a typical injection volume of 20 $\mu$l into a final volume of 0.5 ml. Data were collected every 0.1 s until at least 5 min after consumption reached a plateau. All kinetic data were analyzed by IGOR Pro 6.3 software (WaveMetrics) using user-defined functions. Each oxygen consumption trace was first normalized to correct for calibration error by setting the lowest oxygen concentration to zero. Normalized data were then smoothed using a preset binomial algorithm. The processed oxygen consumption traces were differentiated and fit to user-defined functions (1–3). The first 20-s post-injection was excluded from the analysis because of mixing artifacts. Although the exact instrument response time is unknown, this type of oxygen electrode typically has a response time of the order of

1 s, as such it is sufficiently fast as to not have an impact on the kinetic modeling.

The equations used to fit the data were derived from combining Michaelis–Menten enzyme kinetics with cooperativity of oxygen binding (Equation (1)) with either a one- or two-step activation process modeled based on (pseudo) first-order steps and assuming that only the final state was active, that is, A → B or A → B → C (Equations (2) and (3)). Combining these gives Equations (4) and (5) for the activity of Ero1 as a function of time and [$O_2$]. For activity with ERp72 and for the activity of the C166A mutant of Ero1$\alpha$, there was an apparent loss of activity as a function of time. To model this, we again used a two-step process A → B → C, where A represents the initial inactive state, B represents the activated state, and conversion to an inactive state C represents the loss of activity (Equation (6)). Combining this with Equation (1) gave Equation (7).

$$\text{rate} = V_{\max}*[S]^n / \left([S]^n + K_M^n\right), \tag{1}$$

$$[B] = [A]_0 * (1 - \exp(-k_1 t)), \tag{2}$$

$$[C] = [A]_0 * (k_2*(1 - \exp(-k_1*t)) - k_1*(1 - \exp(-k_2*t))) / (k_2 - k_1), \tag{3}$$

$$\text{rate} = (1 - \exp(-k_1*t))*V_{\max}*[O_2]^n / \left([O_2]^n + K_M^n\right), \tag{4}$$

$$\text{rate} = ((k_2*(1 - \exp(-k_1*t)) - k_1*(1 - \exp(-k_2*t))) / (k_2 - k_1)) \\ *V_{\max}*[O_2]^n / \left([O_2]^n + K_M^n\right), \tag{5}$$

$$[B] = [A]_0 * ((k_1*(\exp(-k_1*t) - \exp(-k_2*t))) / (k_2 - k_1)), \tag{6}$$

$$\text{rate} = ((k_1*(\exp(-k_1*t) - \exp(-k_3*t))) / (k_3 - k_1)) \\ *V_{\max}*[O_2]^n / \left([O_2]^n + K_M^n\right). \tag{7}$$

For fitting the data, we also took into account the background rate of oxygen consumption by the electrode, which we fitted as proportional to [$O_2$] with the background rate of oxygen consumption by the electrode at saturated oxygen level being defined individually for each reaction by a linear fit over 60 s during the pre-injection phase. For the activated wild-type Ero1 with saturating PDI, this correction was maximally 5% of the total rate of oxygen consumption, but the correction became more significant with low activity states. This gave the final equations fitted to as follows:

$$f([O_2], t) = (x*[O_2]) + (1 - \exp(-k_1*(t - t_i)))*V_{\max}*[O_2]^n / \left([O_2]^n + K_M^n\right), \tag{8}$$

$$f([O_2], t) = (x*[O_2]) + ((k_2*(1 - \exp(-k_1*(t - t_i))) - k_1*(1 - \exp(-k_2 \\ *(t - t_i)))) / (k_2 - k_1))*V_{\max}*[O_2]^n / \left([O_2]^n + K_M^n\right), \tag{9}$$

$$f([O_2], t) = (x*[O_2]) + ((k_1*(\exp(-k_1*(t - t_i)) - \exp(-k_3 \\ *(t - t_i)))) / (k_3 - k_1))*V_{\max}*[O_2]^n / \left([O_2]^n + K_M^n\right). \tag{10}$$

All three functions include a parameter, $x$, multiplied by available oxygen, to account for background rate at any given time point that

is a function of available oxygen. Other parameters $k_1$ and $k_2$ are the activation rate constants, $k_3$ is the inactivation rate constant, $t$ is the time modified by injection time $t_i$, $V_{max}$ is the maximal rate, $K_M$ is the Michaelis constant, and $n$ is the Hill constant.

The simplest function that gave a good fit to the data and had random residuals was chosen for each analysis.

Plots of $k_{cat}$, activation rate constants, or halftime of activation versus [PDI] were fit to a single (pseudo) first-order process.

### Gel-based activity assay

Reactions were initiated by mixing 1 $\mu$M of Ero1–PDI heterodimer with substrate solution containing 3 $\mu$M of PDI3FLAG, 10 mM GSH, 0.5 U yeast glutathione reductase (Sigma-Aldrich), 1 mM NADPH, and 2 mM EDTA in activity assay buffer. Reaction was quenched at constant intervals with 50 mM NEM in nonreducing SDS–PAGE loading buffer, immediately heated at +95°C for 5 min, and visualized on an SDS–PAGE gel with Coomassie Brilliant Blue staining. Gel-based kinetics obtained from densitometric analysis were fitted to two-step process.

## Supplementary Information

## Acknowledgements

This work was supported by the Academy of Finland (grants 266457 and 272573), Sigrid Juselius Foundation, The Finnish Cultural Foundation, and Biocenter Oulu. The use of the facilities and expertise of the Biocenter Oulu core facilities, a member of Biocenter Finland, is gratefully acknowledged.

### Author Contributions

A Moilanen: data curation, formal analysis, investigation, funding acquisition, visualization, methodology, and writing—original draft, review, and editing.
K Korhonen: investigation, methodology, and writing—review and editing.
MJ Saaranen: investigation, methodology, and writing—review and editing.
LW Ruddock: conceptualization, formal analysis, supervision, funding acquisition, and writing—original draft, review, and editing.

### Conflict of Interest Statement

A patent for the production system used to make the protein for structural studies using sulfhydryl oxidases in the cytoplasm of *E. coli* is held by the University of Oulu: Method for producing natively folded proteins in a prokaryotic host (Patent number: 9238817; date of patent 19 January 2016). Inventor: LW Ruddock. The remaining authors declare that they have no conflict of interest.

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
