## [Reviewer comments · Life Science Alliance]

Molecular analysis of human Ero1 reveals novel regulatory mechanisms for oxidative protein folding

Antti Moilanen, Kati Korhonen, Mirva J Saaranen & Lloyd W Ruddock

DOI: 10.26508/lsa.201800090

Review timeline:

Submission Date:	11 May 2018
Editorial Decision:	11 May 2018
Revision Received:	4 June 2018
Editorial Decision:	6 June 2018
Accepted:	18 June 2018

Report:

(Note: Letters and reports are not edited. The original formatting of letters and referee reports may not be reflected in this compilation.)

Please note that the manuscript was previously reviewed at another journal and the reports were taken into account in inviting a revision for publication at *Life Science Alliance* prior to submission to *Life Science Alliance*.

1st Editorial Decision

11 May 2018

Dear Dr. Ruddock,

Thank you for transferring your manuscript entitled "Molecular analysis of human Ero1 reveals novel regulatory mechanisms for oxidative protein folding" to Life Science Alliance. The manuscript was assessed by expert reviewers at another journal and the reports have been confidentially transferred to us by the journal editors.

Based on the reports you've obtained elsewhere, we would like to publish your work in Life Science Alliance, pending satisfactory minor revision. As outlined to you prior to submission to our journal, we would expect a point-by-point response to the concerns raised and accordingly text changes. Furthermore, we think it is important to address the request for an additional control (reviewer 1, point 1), and to clarify reviewer #3's comment on the stability of the ERO1-PDI complex. We appreciate that you already outlined how you would proceed to address these issues prior to submitting your manuscript to our journal, and we would like to invite you to proceed as outlined and discussed, and to provide a revised version of your manuscript.

-- High-resolution figure, supplementary figure and video files uploaded as individual files: See our detailed guidelines for preparing your production-ready images, <http://life-science-alliance.org/authorguide>

B. MANUSCRIPT ORGANIZATION AND FORMATTING:

Full guidelines are available on our Instructions for Authors page, <http://life-science-alliance.org/authorguide>

Thank you for this interesting contribution to Life Science Alliance. We are looking forward to receiving your revised manuscript.

1st Revision – authors' response

4 June 2018

We thank the referees for their hard work in evaluating the original manuscript and are pleased that our work “provides important new mechanistic insights into Ero1 function and regulation”, that “Either of these findings merits a full paper” and “overall the conclusions are well-supported by the data.” We have made substantive changes based on their comments indicated in red in the annotated version of the revision (changes from EV to supplementary figures and other stylistic changes for the journal not annotated) and feel that the manuscript is much improved due to the referees input. In specific response to the referee’s comments:

Referee #1:

Major issues:

1. Throughout a series of the in vitro experiments, they used Ero1-alpha and Ero1-beta proteins produced by their original expression systems and purified by combinatorial chromatography. Thus, they assert their preparation and usage of 'homogeneously folded and disulfide bonded' Ero1 proteins. However, only the CBB stained gel and the electron spray ionization analysis (data not shown) are not sufficient to prove the homogeneity of their Ero1 preparations. To demonstrate the homogenous disulfide bond pattern of their Ero1 preparations, other experimental data such as trypsin digestion followed by LC/MS are strongly required.

We agree with the reviewer that the wording of the state of our Ero1 needs careful consideration. However, we were very careful to avoid saying our Ero1 was homogeneously folded. Using our CyDisCo system for generating disulfide bond containing proteins in the cytoplasm with very few exceptions we generate “homogeneously folded and disulfide bonded proteins” (based on both published and a very large body of unpublished data from >300 proteins and including published and unpublished x-ray structure determination).

However, we have no direct evidence that our Ero1 is “homogenous” and so we did not claim it was. We have now removed the word “homogeneously” from our generic description of the production system. We do have SDS-PAGE that shows an OX2 like state for both Ero1 and ESI-MS, both of which imply a high degree of homogeneity. We have preliminary data from trypsin digestion with MS, but the issue with this methodology is two-fold. Firstly, we do not believe that

this method can demonstrate the degree of homogeneity. Secondly, the mixed disulfide state of Ero1-PDI breaks down due to nucleophilic attack by C400 of PDI on the mixed disulfide giving rise to a reactive free thiol and subsequent thiol-disulfide rearrangement i.e. we have the same issue that we have in trapping the mixed disulfide state by NEM for SDS-PAGE analysis. This kinetic partitioning invalidates the approach.

Before submission, we had some rpHPLC data on the homogeneity of the proteins. We did not include this in the original submission as we only had the data for Ero1□, the Ero1□ analyzed was produced under slightly different experimental conditions (prior to optimization) and we felt that the data did not add anything to the manuscript. It is clear now that the addition of the degree of heterogeneity/homogeneity requires additional data to support and so we have done rpHPLC of the two complexes both as purified and under reducing conditions. PDI has only short-range disulfides and shows only a small shift in elution position upon reduction. In contrast, Ero1 has long-range disulfides and so shifts substantially upon reduction. A comparison of the symmetry of the Ero1 and PDI peaks along with the shift in the Ero1 peak position upon addition of DTT, indicates that the Ero1□ produced is predominantly a single species and oxidized (Fig S2A). Ero1□ is less homogenous, but the majority (78%) is in the oxidized state (Fig S2B). We have also added details of the mass spectrometry results as Tables S1 and S2.

2. In Fig 1, the authors showed the formation of disulfide-linked human Ero1-PDI complex via Cys166. The authors should describe the exact conditions for the co-purification, including incubation time after mixing, NEM concentration and redox state of PDI used. Lane number needs to be added to all gel data. As to panels C, D, E and F, the gel data under non-reducing condition also need to be displayed like in panel B, which would give important information on the yields of the covalently linked Ero1-PDI complexes.

The Ero1-PDI complex is formed *in vivo*, there is no mixing of purified components and so no incubation time, no need for NEM and the redox state of PDI that forms the complex is unknown. The complex co-purifies through all of the purification steps for Ero1. We have amended the supplementary methods section to further clarify this and added the fact that PDI is co-expressed to the results section to aid the reader.

We feel that numbering all of the gel lanes and hence requiring the reader to cross check between the figure and legend is less user friendly than the current method for labelling gel lanes with what they contain.

We feel that showing full gels reducing and non-reducing, in the style of panel B for panels C, D, E, and F would not add any relevant information (panel E has the two halves covering both gels) while more than doubling the size of the figure. Most of the gels are included to show that a stable co-purifying complex is formed *in vivo* with the various cysteine mutations and they demonstrate clearly which result in co-purification and which do not.

3. To emphasize the functional importance of Cys166/Cys165, the authors should address how highly conserved Cys166/Cys165 is among the Ero1 family enzymes.

We have added a statement on conservation of this Cys among the 451 reported sequences for Ero1 from chordates to the manuscript. We limited ourselves to chordates as from other ongoing projects we are aware of a small number of Ero1 family members from other organisms which are substantially different in terms of regulation and whose PDI-family members are likewise substantially different.

4. Although their analysis of the lag time in the complete time-course of Ero1 oxygen consumption trace suggests the presence of a two-step activation process for Ero1-alpha, the process of drawing this conclusion is very unclear. Given that the activation process involves the complex formation with PDI, did the complex-breaking mutant C166A show a single-step activation process? What do the red and black lines in Figure 4B actually indicate, respectively? After all, exact mechanisms of each activation step in Ero1-alpha is not addressed in this paper.

The determination of whether the activation process was one or two step was based on the kinetic analysis undertaken and which model gave the best fit to the data and random residuals. When multiple models gave equivalent fits the simpler one was adopted. A sentence on this has been

added to the supplementary methods. For wild-type Ero1 \square a two-step activation model was always observed with PDI as a substrate. For wild-type Ero1 \square , a two-step activation model was observed at low [PDI], but one of these steps became unmeasurably fast at high [PDI]. Minor changes have been made to the manuscript to help clarify this.

As stated in the manuscript the Ero1 \square “C166A mutant showed enhanced activation rates” and “fitted best to a model with a single activation step”.

The two curves in Fig 4B represent the dependence on [PDI] of the observed first order rate constants for Ero1 α activation. In the revised manuscript, the red symbols have been replaced, as the use of colour does not otherwise enhance the figure.

5. Related to the above comment, Figure 5B is very unclear and large parts of their model do not seem to be based on experimental data. The residue number of Cys involved should be shown in the figure. Perhaps, there is no experimental evidence for the redox state of PDI during the process of PDI exchange and for which active-site cysteines of PDI are involved in each step of the reactions illustrated in this model.

As presented in the results, e.g. Fig 1, we know that the mixed disulfide is formed between C165/C166 of Ero1 and C397 of PDI. We also know that activation of Ero1 is a two-step process (Fig 4) and shows a dependence on added PDI. The PDI dependence of both steps of the activation of Ero1 \square show an affinity for PDI similar to that of the KM for the substrate. Furthermore, the kinetics of PDI exchange were similar to those of activation. As such, we believe the model is fully supported by the data.

We attempted to dissect out which disulfides in the Ero1-PDI complex were being reduced during activation and in which order. However, as stated many of the mutants either could not be produced or showed structural instability. The C166A mutant showed enhanced activation rates and fitted to a single activation step implying breaking the Ero1-PDI mixed disulfide is an important step in activation. Since the remaining disulfides broken and the order could not be determined unambiguously, we prefer to leave the residue numbers unmarked in Fig 5B.

6. The reason why 'MAMA' mutant was used in Figure 1E should be explained.

Historically the active sites of redox enzymes have either been mutated to Ser or Ala to inactivate them. Serine has the advantage that it is closer in size to Cysteine and has a hydrophilic side chain. However, serine is a strong regular structure breaker and we have a substantial body of unpublished data that mutating the active site cysteines of PDI family members to serine causes structural changes. Alanine lacks this issue, but it is smaller and aliphatic and so may cause other local structural issues.

Methionine is less widely used, but in our hands the replacement of the N-terminal active site cysteine with a methionine is well tolerated by human PDI family members and retains a (unreactive) sulfur atom at a near equivalent position.

The C-terminal active site cysteine is buried and the larger methionine is not tolerated at this position, a serine at this position causes disruption of the helix and an alanine is the least disruptive choice.

The data for this summary spans more than 2 decades, draws on multiple PDI family members, multiple projects and substantial parts of the results have not been reproduced sufficient times for publication (but the sum of whole gives a clear indication).

Given the complexity of the reasoning we would prefer not to add any clarification to the manuscript as we feel it would detract from the main message. These response to reviewers will be available online should this manuscript be accepted and we hope this is sufficient.

7. The regulatory role of the Cys208-Cys241 disulfide has also been investigated biochemically in Kanemura et al. JBC (2016). This paper should be cited.

We agree with the referee that this important paper should have been included in our manuscript. It has been added in the revised version.

8. page 3, line 9; "Cys81-Cys391" should read "Cys85-Cys391".

This has been modified.

9. page 11, line 4; Figure 4E" should read "Figure 4F"

This has been modified

10. A hyphen is missing at many sites of the text. There are a number of careless grammatical errors in the text. Careful rewriting is required.

We have carefully gone through the manuscript before the resubmission making all corrections we could find

Referee #2:

Major points:

- Major parts of the study are based on the observed interactions between PDIs/Ero1 from recombinant production in bacteria, without any flanking experiments in mammalian cells. This does not allow to judge the biological significance of many findings reported in the study.
- Ero1alpha/beta have several N-glycosylation sites, which have been shown to become modified in mammalian cells. These would remain unmodified in *E. coli* and may affect function/PDI interactions. This is of particular relevance, since a lot of the conclusions from the study are based on comparisons of in vitro kinetic parameters.

The majority of the literature on the molecular mechanisms of action of Ero1 are based primarily on in vitro analysis of protein produced in *E. coli* and refolded and/or reoxidized in vitro. As such our work is close to the norm in the field, except we use in vivo folded protein rather than in vitro refolded material and we use the wild-type enzyme rather than the usual C166A/C165A mutants. We are unaware of any published data that suggests a role for the N-glycans in Ero1 playing a functional role, but it is possible that they do and that nearly 20 years of molecular enzymology on Ero1 is wrong. Given the complexity of multiple oxidative folding pathways in mammalian cells it is unclear to us how detailed mechanisms on a single enzyme could be elucidated in vivo. The existence of the complex we described is supported by previous in vivo based publications that are cited in the manuscript (Benham et al, 2000; Appenzeller-Herzog et al, 2008).

- Ero1beta could not be purified without as the C165A mutant. This is surprising and may argue for the fact that PDI actually recognizes an incompletely folded Ero1beta from *E. coli* as a substrate. Along the same lines, it is e.g. stated that Erp57 needed to be co-expressed to avoid incorrect folding of Ero1. The reason for this remains unclear to this reviewer, as no reference is given for this, and may again point towards unstable folding of Ero1 in *E. coli* and its substrate-like recognition by PDI. Furthermore "Ero1alpha C166A showed a time-dependent loss of activity", also arguing for structural instability and raising concern about the nature of the PDI/Ero1 interaction.

It is unclear to us why it is surprising that the C165A mutant of Ero1 was unstable given one of the key findings of the manuscript is that PDI and Ero1 form a stable complex and that this is mediated by C165 (nearly all proteins in heterodimeric complexes are less stable when either expressed without their interaction partner or in a mutated form such that the interaction is destabilized). A similar argument holds true for the Ero1 C166A mutant. While we do not have the raw data from all previous publications from other groups, the time-dependent loss of activity of the C166A mutant appears to be visible in all, it is just not commented on fully.

It is possible that the Ero1 we have produced is incompletely folded (see also referee 1 comment 1), but all of the indications are that our material is more homogeneous and has higher biological activity, especially towards endogenous substrates, than any refolded material reported to date.

The requirement for ERp57 co-expression when mutant PDI was used is based on our observations, not on anything previously published. We have amended the text to clarify to the non-specialist that ERp57 is a protein disulfide isomerase and that the mutations in the active site of PDI will decrease the isomerase activity of PDI.

Minor points:

- In the introduction, the authors state that any structural studies on the "true nature" of PDI-Ero1 heterodimers are missing. It is not clear to this reviewer what is meant by this expression.

The original statement was ambiguous and has been amended.

- In the introduction, the authors cite turnover numbers for Ero1 of ca. 0.3 s⁻¹, which are "inconsistent with the fast in vivo oxidation rates". What is the latter statement based on? Typical protein folding reactions in the ER take from seconds to minutes.

While the time taken for most proteins to reach the native state in vivo is usually minutes to hours (the fastest folding is limited by the rate of protein translation i.e. circa 30 seconds per 10kDa in eukaryotic systems), the majority of available data suggests that the rate limiting step in disulfide bond formation is late stage isomerization events in intermediates with quasi-native structure and not oxidation. This, combined with the typical number of disulfides in folding proteins and the relative concentration of Ero1, suggests that a faster turnover than previously reported is required if Ero1 is to form the primary route for disulfide bond formation. An additional complexity is that under redox conditions mimicking those in the ER much of the Ero1 is in an inactive state rather than an active state and turnover can drop by over 90% under conditions mimicking typical ER redox states. This apparent paradox in rates has previously been discussed in the literature and an appropriate citation has been added to the revised manuscript.

- In the introduction the authors claim that their study is "the first biochemical evidence for the mechanisms of how cells trade the need to maintain disulfide bond formation at low oxygen concentrations"; this is maybe a bit of an overstatement taking into account i) the lack of cellular data (see above) and ii) previous studies, e.g. Koritzinsky et al., JCB 2013: Two phases of disulfide bond formation have differing requirements for oxygen

We contend that this manuscript provides the first biochemical evidence of how it is done, i.e. the mechanisms. We are aware of the Koritzinsky et al publication, which has sparked a lot of interesting discussion in the field. However, there are multiple routes for disulfide bond formation in vivo, both protein based (eg Ero1, Qsox, VKOR) and mediated by low molecular weight species (eg glutathione, peroxide, DHA, ROS etc) and due to the complexities of in vivo systems Koritzinsky and coworkers were unable to elucidate which routes were active under their different experimental conditions. As such we feel that adding a citation would require a large additional section on the different routes to be inserted in the text which would distract from the message of this manuscript. If required we can add a section on the data in Koritzinsky et al to our manuscript.

- It is very difficult to conclude from the results section at which point which ER oxidoreductases where co-expressed (and for which purpose) with Ero1.

The system for making disulfide bonded proteins has PDI as a catalyst of isomerization. This is present in all expression tests – except where it is stated that mutant PDI is used instead. When a mutant of PDI was used (which compromises its isomerase activity) an alternative member of the PDI-family, ERp57, was also co-expressed as the alternative isomerase. We have amended the opening paragraph to clarify the PDI co-expression (see above regarding ERp57 related amendment).

- The authors fail to cite/compare their findings on oxygen-level dependent regulation of Ero1 to oxygen levels in cells - which several recent studies have measured.

There are a large number of papers on oxygen levels in biological systems and it is unclear to which the referee is referring. Ultimately what we would need to make correlations would be what is the oxygen concentration inside the ER (oxygen gradients exist within cells), inside cells in a living

organism (oxygen gradients within tissues and dependence on physiology further complicate this). We are unaware of any published data on this and what data is available is often contradictory or controversial. To our knowledge there is not even a consensus in the field on what oxygen levels constitutes “hypoxia”, but the range is above our determined KM for oxygen for both human Ero. Similarly there is no consensus in the field on what constitutes “hyperhypoxic” or “severe hypoxic” conditions, but the range is below our determined KM for oxygen.

Given the differing views in the field, any discussion on this would have to be extensive and we feel this could be a distraction from the main message in this manuscript. Accordingly we would prefer to not add a section discussing this.

Referee #3:

Regarding the part about Ero1-PDI interactions, the authors are correct in saying that cysteines 166/165 of human Ero1 enzymes are often either eliminated or ignored in biochemical studies. For structural studies of Ero1, it was reasonable that this cysteine be removed for the practical purpose of avoiding unwanted disulfide-mediated dimerization of protein stock solutions. However, it is indeed a shame that this cysteine got to be labeled a "non-functional" cysteine in the literature. The simple fact of its availability for (unwanted) reactions would make it suspect for possible functional purposes! It is useful that the authors reverse the record on that point. However, the section at the beginning of the discussion that mentions "in vitro refolding" does not seem warranted. The structural studies on Ero1 and many subsequent biochemical studies were not done on in vitro refolded protein. Producing human Ero1 proteins in the soluble fraction of E. coli lysates does not seem to be the real novelty of the current manuscript. The whole first paragraph of the results should thus be condensed, and the focus should be on the mixed disulfide complex formed between Ero1 and PDI in their particular expression system.

We agree with the referee that the overall stress on this issue in particular in the first paragraph of the discussion is inappropriate and have amended it accordingly. The shortening of this section allows us to expand other sections where further clarification is needed without increasing the overall length of the manuscript. However, we feel that the opening paragraph of the results is essential to set the scene for both specialists and non-specialists.

The next issue that arises in the manuscript is that of the stability of the Ero1-PDI complex. The results of PDI exchange experiments are reported, and from the methods section it is evident that the experiments were conducted for an hour at room temperature. They also report that "oxidized PDI molecules were exchanging in the complex." If this is the case, then there is no requirement for nucleophilic attack on the complex by exogenous reduced PDI. Instead, and consistent with the findings regarding PDI residue C400, the complex would need to have the capacity to disassemble (even in the native states of the proteins) at some rate by attack of PDI C400 on the mixed disulfide between PDI C397 and Ero1 C166, liberating oxidized PDI and leaving Ero1 C166 reduced. Then Ero1 C166 could attack a C397-C400 disulfide in a new PDI molecule and form a new complex. According to this exchange mechanism, purified complex would be expected to spontaneously disassemble, which would be particularly evident if the complex were diluted to lower the rate of reformation upon disassembly. It would be good if this issue could be clarified experimentally and textually.

During the six years we have worked on this complex we have found no evidence that the inactive Ero1-PDI complex spontaneously disassembles on a physiologically relevant timescale unless exogenous PDI is present. The data that the [PDI] dependence for activation is similar to the apparent KM during catalysis suggests that the activation proceeds via exchange of exogenous reduced PDI with PDI from the complex with both the incoming and outgoing PDI competing for the same site in Ero1 i.e. the β hairpin. Based on this and the stability of the complex in the absence of exogenous PDI, our working hypothesis is that the PDI in the complex is interacting with the Ero1 via both its b' domain (the α -hairpin) and its a' domain (the C397 mixed disulfide with C166/C165). We assume that there is relatively slow dynamic thiol-disulfide exchange between the mixed disulfide state and the oxidized PDI state – due to nucleophilic attack by C400, but that the loss of the mixed disulfide state does not release PDI from the complex due to the interaction with the α -hairpin. It is only when both interactions are broken simultaneously – something that requires

exogenous PDI exchange or denaturation – that the PDI is released from the complex. This gives rise to the unexpected result that both oxidized and reduced exogenous PDI can exchange with the PDI in the complex. The former results in no net change in the complex, while the latter leads to activation of Ero1. This dual interaction (mixed disulfide and \square -hairpin) also explains why PDI family members such as ERp57, ERp72 and P5 are inefficient at activating PDI despite having WCGHC active sites which should be able to displace PDI from the mixed disulfide state i.e. they lack the interaction with the \square -hairpin. To be able to confirm this we would need to solve the structure of the PDI-Ero1 complex, but to date it has proved intransigent to crystallization, or we need to observe the transient tertiary complex formed by the ERo1-PDI complex with exogenous PDI during the exchange. This complex is so transient we have not been able to observe it. Within the limitations imposed by this, we have modified the discussion and added a supplementary figure (S13) to elaborate on this.

An issue similar to the "refolding" objection above arises in the enzyme kinetics section. The important observation is the activity of the authors' complexes and the subsequent findings based on them. Whether or not others have reported k_{cat} or K_M values is a distraction. In fact, rather than focusing on the authors' interesting findings at this point, this reviewer was sent to the literature to find reports of k_{cat} and K_M measurements of human Ero1 for substrates. After readily finding in Blaise et al., JBC (2010) reports of turnover numbers and K_M of human Ero1alpha for the (presumably non-physiological) substrate thioredoxin, this reviewer decided to resist the continued temptation to search for more and instead recommend that this paragraph, like the first paragraph of the results, be condensed and focused on the important point. That point is describing the system used to make the oxygen consumption measurements (i.e., wild-type Ero1-PDI complex, GSH, etc.).

As far as we are aware there is very limited data in the literature on the k_{cat} or K_M values for Ero1 using physiological substrates i.e. PDI family members (we cited both references we were aware of). Given we have a new production system feel it is essential to cross compare the activity we obtain with that previously reported. In particular the fact that our material appears to be more active than any previously reported suggests that our material is correctly folded (see objections referee 2 regarding the importance of this point). Since we had no access to the raw kinetic data we estimated values from previous publications based on tangents drawn to the oxygen consumption traces. We are reluctant to show these values as they are only estimates.

We agree with the reviewer that the wording in our previous submission would have readers scrambling to find previously published data and so we have amended the section, but given the concerns regarding the question of how folded our material is, we have kept points regarding relative activity.

One technical issue to perhaps consider in this work is that of the response time of the oxygen electrode. The time-courses of the experiments shown in the manuscript are long, but the changes in the rates of oxygen consumption at low oxygen concentrations occur over a short time window. The Ero1 control is useful in this regard, but it might be good to know the relevant numbers and how big of an issue the instrument response time is in the kinetic modeling.

We agree with the referee that ideally we would have precise numbers for the instrument time, but unfortunately these are not available. This type of oxygen electrode has a response time of the order of 1s. The manufacturer states that the 10-90% response time is <5 seconds. We have added a sentence to the methods to this effect.

While the changes in rates of oxygen consumption at low oxygen concentrations appear to occur over a short time window, this is not the case. The high hill coefficient results in a very drastic drop in Ero1 activity below the K_M and so below the K_M oxygen consumption is slow. For example, the time taken to go from 5 to 1 \square M oxygen for the Ero1 \square complex trace shown in Figure 2 is circa 300s and in no experiment is it less than 3 minutes. Given this we feel that the instrument response time is sufficiently fast as to not have an impact on the kinetic modeling.

In the discussion, this reviewer felt that two points were missing. One is a clearer explanation of why cooperativity in oxygen utilization by Ero1 is beneficial. The authors do state, "Ero1 is fine tuned to utilize oxygen as a terminal acceptor in a manner that is non-deleterious to the cell." But

could they explain what exactly would be deleterious about using oxygen non-cooperatively? It may be obvious, but it would still be useful to state explicitly.

A short addition has been made to the discussion to clarify this.

Also, presumably the cooperativity is due to some unique mechanism. It is admittedly beyond the scope of this manuscript to reveal the details of this mechanism, but at least some mention of the interest in uncovering this mechanism would be nice. Cooperativity in oxygen binding is obviously a very well-known concept in biochemistry, but thinking about how Ero1 might accomplish cooperativity in oxygen utilization leads us away from, rather than toward, familiar systems like hemoglobin. (Ero1 is not a tetramer...) This reviewer found some discussion in the literature about oxygen utilization by sulfhydryl oxidases including an examination of oxygen channels in enzymes that use oxygen efficiently vs. those that don't. In addition, much has been written about how oxygen is activated for use by flavin-dependent oxidases. But little seems to be available regarding cooperativity in oxygen utilization by oxidases. This point at least seems to deserve a mention to strengthen the novelty of the authors' findings and point the direction for future work.

We fully agree with the reviewer that this is an important point. We discussed this extensively before the original submission and felt that the inclusion of details on oxygen channels from the literature would unduly lengthen the discussion without adding to the essential question – what are the mechanisms by which cooperativity arises? During the writing of the original manuscript, we became aware of systems consisting of monomeric enzymes which have a single substrate-binding site but which demonstrate allosteric binding or activation. The best characterized of these we found is glucokinase and our suspicion is that a similar mechanism based on order-disorder transitions gives rise to the cooperativity seen in Ero1. Since we have no evidence for this we have added only 4 sentences to the discussion, to clarify to the reader that the effect probably arises from intra- rather than inter-molecular mechanisms and to raise the awareness of allosteric activation mechanisms in monomeric single substrate binding site enzymes.

We hope that with these changes the manuscript will be suitable for publication in Life Science Alliance.

2nd Editorial Decision

6 June 2018

Thank you for submitting your revised manuscript entitled "Molecular analysis of human Ero1 reveals novel regulatory mechanisms for oxidative protein folding".

We appreciate the way you addressed the reviewer comments both by careful text changes and by inclusion of additional experiments.

We would thus be happy to publish your paper in Life Science Alliance, congratulations on this nice work!